# Analysis of Factors Affecting Neutralizing Antibody Production after COVID-19 Vaccination Using Newly Developed Rapid Point-of-Care Test

**DOI:** 10.3390/diagnostics12081924

**Published:** 2022-08-09

**Authors:** Hyeon Woo Shim, Jae hang Shin, Shang Cheol Shin, Hwa Jung Lee, Kyung Soon So, So Young Lee, Jae Woo Jun, Jeong Ku Seo, Hwa Seop Lee, Suk Young Lee, Seung Hyun Kim, Sun Jong Kim, Kyoung-Chol Kim, Gyu Ha Ryu

**Affiliations:** 1BioFront, HanGang Xi Tower, 401 Yangcheon-ro, Gangseo-Gu, Seoul 07258, Korea; 2EONE-DIAGNOMICS Genome Center Co., Ltd., #291, Harmony-ro, Yeonsu-Gu, Incheon 22014, Korea; 3Department of Medical Device Management and Research, SAIHST, Sungkyunkwan University, Seoul 06355, Korea; 4Major Clinic #452, Dogok-ro, Gangnam-Gu, Seoul 06279, Korea; 5DiaVison, 32 Maeheon-ro 16-gil, Seocho-Gu, Seoul 06770, Korea; 6Office of Technology Commercialization, Samsung Medical Center, Seoul 06351, Korea

**Keywords:** POCT, S1 RBD IgG antibody, neutralizing antibody, vaccine

## Abstract

(1) Objective: To investigate the factors that affect rates of neutralizing antibody production and duration after vaccination using the newly developed SARS-CoV-2 POCT. (2) Methods: The production of immunoglobulin and neutralizing antibody in clinical subjects who completed various vaccines was analyzed using the POCT, the semi-quantitative was interpreted by measurement application, and the quantified neutralizing antibody titers were using the ELISA. (3) Results: According to the clinical performance analysis of the POCT, the clinical sensitivity and the specificity were 96.8% (90/93) and 97.7% (167/171), respectively, for the S1 RBD IgG antibody. The clinical sensitivity was 92.22% (83/90), and the clinical specificity was 100.00% (174/174) for neutralizing antibodies. Factors influencing antibody production were analyzed using the whole blood of the five types of second-completed vaccinators (N = 736, 20–80 years old). General and neutralizing antibody and showed significant differences in age (*p* < 0.0001), vaccine type (*p* < 0.0001), inoculation interval (*p* < 0.0001), pain score (*p* < 0.0001), diabetes (*p* < 0.0001), and hypertension (*p* = 0.002). The gender (*p* = 0.021) and chronic fatigue (*p* = 0.02) did not show the significance. (4) Conclusions: An acquisition of immunoglobulin and neutralizing antibody varies according to vaccine type, age, days after vaccination, pain degree after vaccination, and underlying diseases. The POCT used in this study will be utilized for clinical recommendations such as deciding whether to receive additional vaccines through the immediate rapid determination of neutralizing antibody generation in the clinical site.

## 1. Introduction

The coronavirus outbreak began in December 2019 with a mass outbreak of pneumonia in Wuhan, Hubei Province, China. The World Health Organization (WHO) identified the cause of pneumonia as a novel coronavirus similar to SARS and MERS, named “SARS-CoV-2” by WHO. Symptoms of the infection include fever, dry cough, and shortness of breath, and it has been reported that worsening symptoms can lead to pneumonia, kidney failure, and death in severe cases [1,2]. Recently, emergency clinical trials have been approved and implemented following vaccine development. Still, it has been reported that each vaccine used has a different effect on neutralizing antibody production. The vaccine currently being developed and used is known to have a preventive effect of about 52.95% against the wild type of the virus [3]. In addition, it is reported to have effects such as reducing specific diseases, hospitalization rates, and mortality. However, current research reports on the effect of generating neutralizing antibody and the prevention of infection after vaccination, which can be used as a basis for the development of vaccines specialized for various mutant viruses, are insufficient.

According to a recent study for seven vaccines, the efficiency of the S1 RBD-binding antibody is reported to be greater than 90%, but the efficiency of the neutralizing antibody shows to be less than 70% [4]. Therefore, despite various research and development efforts necessary for COVID-19 treatment and prevention, the eradication of COVID-19 is considered impossible in a short period. For this, it is deemed essential to develop a technology that can easily and quickly read or measure the generation and duration of the neutralizing antibody according to individuals or populations related to infection prevention after vaccination [4,5,6].

In general, whether antibodies are produced by an infection or vaccine is measured and read using enzyme-linked immunosorbent assay (ELISA) or point-of-care test (POCT) lateral flow assays using blood samples. The recently developed SARS-CoV-2 immunoglobulin M (IgM)/immunoglobulin G (IgG) antibody test can be utilized to confirm the production of antibodies (IgM) at the initial stage of infection and antibodies mainly involved in immune responses (IgG) after coronavirus infection [7]. However, it is known that this test is not appropriate for the actual determination of the formation of neutralizing antibodies [8].

Recently, it has been reported that S1 RBD IgG-specific antibody production is closely related to the neutralizing antibody. It has been reported that re-infection after a mutated SARS-CoV-2 outbreak precludes the neutralizing antibody reactions due to changes in the energetics between the mutated RBD and the neutralizing antibody of RBD protein mutation [9,10,11]. For most vaccines, the ability to neutralize specific mutants is significantly reduced, and the development of a rapid diagnostic test for a neutralizing antibody of various mutants may be an important basis for the study of the production of the neutralizing antibody for new mutants [12]. In addition, studies on the immunological responses of various groups of age and gender and genetic factors by vaccines and clinical associations are also considered significant [13].

SAR-CoV-2 was fused with the host by mediating angiotensin-converting enzyme 2 (ACE2) as the primary receptor, and it was confirmed that infection through “pulmonary epithelial cells” of the lungs in the respiratory tract, where the receptor was most distributed, was the main route of transmission [14,15,16]. In addition, the antigenic protein of the virus mediated by ACE2 acting as a receptor for SARS-CoV-2 is known as the spike (S) protein. The S protein consists of subunits 1 (S1) and 2 (S2). S1 is known to be involved in host cell binding, S2 is involved in the fusion of cell membranes and viruses, and antibodies that inhibit this specific binding are known as neutralizing antibody [15,16,17]. According to recent SARS-CoV-2 research, IgG antibodies binding to N proteins generally do not function as direct neutralizing antibodies, and the antibodies binding to S proteins or specific parts (RBD) of S proteins function as neutralizing antibodies [18,19].

Standard tests for measuring neutralizing antibody production rates of various viruses are known as the plaque-reduction neutralization test (PRNT) and SARS-CoV-2 microneutralization test. These tests have limitations, such as acquiring biosafety level three (BSL3) culture facilities and trained researchers. Recently, the FDA approved the ELISA test to measure chromatic change using the ACE2-RBD competitive binding principle as a neutralizing antibody measurement method [20], which has been proven by many research reports on COVID-19-neutralizing antibody production. The SARS-CoV-2 virus emerged at the end of 2019; COVID-19 neutralizing antibody tests were developed for only about a year, and research is still underway on the characteristics of COVID-19-neutralizing antibody tests. The neutralizing antibody can be used as an auxiliary diagnostic method for inflammation to determine the infection rate or level of herd immunity in a population and the immune status of patients after infection/vaccination [21,22,23].

Currently, SARS-CoV-2 antibody tests are determined mainly to detect IgG/IgM antibodies and neutralizing antibodies. ELISA neutralizing antibody measurement methods still have limitations, such as sample limitations using serum or plasma and the process of transferring samples to the laboratory. Despite the low sensitivity of the rapid immunochromatography test, it has the advantage of convenience, low cost, no loss of reagents even after testing one sample at a time, simple preparation without special equipment, and the ability to obtain results within 15–30 min. Therefore, the development and application of the POCT using the same principle is considered an excellent method of overcoming the limitations of conventional neutralizing antibody measurements [24]. The POCT used in this study, RapiSure^TM^ (EDGC^TM^) COVID-19 S1 RBD IgG/Neutralizing Ab Test, can perform two types of diagnosis in a single cassette. S1 RBD IgG antibody and SARS-CoV-2 neutralizing antibody were simultaneously examined in whole blood, serum, and plasma. The POCT used in this study can be used as a self-measurement method for antibodies produced after infection or vaccination, and it is anticipated to be used as a criterion for determining whether to administer additional doses of vaccines according to immunity by quickly measuring the effect of antibodies produced after infection or vaccination.

## 2. Materials and Methods

### 2.1. Human Samples Collection

In this study, 736 people who completed the secondary vaccination of 5 types of vaccines (AZD1222 corresponds to AstraZeneca COVID-19 vaccine, AstraZeneca, Oxford, UK; BNT162b2 corresponds to BioNTech COVID-19 vaccine, Pfizer, New York, NY, USA; mRNA1273 corresponds to pikeVax COVID-19 Vaccine, Moderna, Cambridge, MA, USA; AZD1222 + BNT162b2, and Ad26COV2.S corresponds to Jcovden COVID-19 Vaccine, Janssen, Beerse, Belgium) were subjected to a retrospective study on the production of S1 RBD IgG antibodies and neutralizing antibody. The study subjects were men and women within the ages of 20–80 who had received the second dose of 1 of 5 types of vaccines examined at least two weeks prior. Those with a history of natural infection in the past, have not been vaccinated, and have been vaccinated within two weeks or two more than six months ago were excluded. This study was conducted after institutional review board approval of Major Clinic (GM-2021-02), and all subjects agreed to and provided the informed consent prior to participating in the research. We excluded the sample from the study if hemolysis or severe lipemia occurred in the sample after collection. Each sample was tested immediately after collection, and if storage was necessary, it was stored at 2–8 °C for 24 h.

### 2.2. Rapid Immunochromatographic Assay

The RapiSure^TM^ (EDGC^TM^) COVID-19 S1 RBD IgG/Neutralizing Ab Test is based on the principle of lateral flow colloidal gold immunoassay for the detection of S1 RBD antibody and neutralizing antibody to SARS-CoV-2 in whole blood, serum, or plasma. In the case the of S1 IgG antibody test, if the sample contains S1 RBD antibody, it forms a complex with the gold-labeled antigen (RBD of the S1 spike antigen). The complex moves forward under the action of chromatography and reacts with the coated antibody (Mouse anti-human IgG monoclonal antibody) at the T band and develops color, indicating a positive result. On the other hand, if the sample does not contain the S1 RBD antibody, no complex will be formed at the T band and no colored band will be shown, indicating a negative result. In the case of neutralizing antibody test, the membrane is pre-coated with Angiotensin I Converting Enzyme 2 (ACE2) on the test line region of the strip. During testing, the blood specimen reacts with S-RBD conjugated colloid gold. The mixture migrates upward on the membrane chromatographically by capillary action to react with ACE2 on the membrane and generate a colored line. The appearance of this colored line indicates a negative result, while its absence or similarity to the R line indicates a positive result. To serve as a procedural control, a colored line will always change from blue to red in the control line region, indicating that the proper specimen volume has been added and membrane wicking has occurred. The sample amount used in the test was read within 15 min using 25 μL for S1 RBD IgG and 50 μL for the neutralizing antibody. After the inspection, it was determined through visual reading. The image was captured using a smartphone camera (Galaxy S20 plus, Samsung, Suwon, Korea), the signal strength of the test and control line was analyzed (Diavision, Seoul, Korea), and the results for visual reading were compared and verified (Appendix A).

### 2.3. Analysis of Results Using the Measurement Application

POCT interpretation was conducted using smartphone image analysis software instead of a visual inspection. After the test was complete, images were captured using a smartphone camera (Galaxy S20 plus, Samsung), and the signal intensity of each line (test, reference, and control) were analyzed using a smartphone-based image analysis software (“SmartVision” Diavision, Inc., New Delhi, India). The software controls automatic exposure level and white balance and focuses for optimal analysis conditions and notice by green guidelines on the screen for optimal angle and distance. Captured images were analyzed by color spectrum and pixel intensity, one by one; all lines, calculated to ratio, compared each line pixel intensities with reference line pixel intensity. The calculation formula is straightforward, T/R ratio (%) = (test line pixel strength/reference line pixel strength) × 100%. The S1 RBD IgG antibody verifies whether the test line signal is present or not.

### 2.4. Rapid Immunochromatographic Clinical Performance Assessment

In order to evaluate the clinical performance of POCT, clinical sensitivity and specificity were evaluated compared to the final confirmation results conducted by RT-PCR and PRNT_90_ (Institutional Review Board 2021GR0481, Korea University Guro Hospital). An evaluation was performed by collecting 264 serum and plasma residual samples suitable for the subject and sample selection criteria. Negative patients have no history of vaccination against SARS-CoV-2 and confirmed negative results for RT-PCR. The presence of neutralizing antibody against SARS-CoV-2 was confirmed using the PRNT_90_ assay. The PRNT_90_ was performed as described previously [25]. Briefly, serum samples were diluted from 1:10 to 1:160 using phosphate buffer saline. Approximately 100 plaque-forming units of SARS-CoV-2 (BetaCoV/Korea/KCDC03/2020, NCCP 43326) were incubated at 37 °C for 1 h. Subsequently, the mix of serum with the virus was overlaid on a VeroE6 monolayer and the plates (NEST Scientific, SPL Life Science, Pochen, Korea) were incubated for 1 h at 37 °C in 5% CO_2_ with intermittent rocking. The monolayer was fixed using 4% PFA, and the plaques were visualized by crystal violet staining. The number of plaques in the wells with no sera was counted and taken as control. Samples with PRNT_90_ titer 1:20 were considered PRNT_90_ positive, and the remaining were PRNT_90_ negative.

### 2.5. Serologic Antibody ELISA Assay

As a control test for neutralizing antibody production, SARS-CoV2 Surrogate Virus Neutralization Test Kit (GenScript, CAT# L00847-A, Piscataway, NJ, USA) was used to compare and verify according to the test indication [26]. The neutralizing antibody production reading criteria were positive if the signal inhibition cut-off value were 30% or greater, and negative if the value was 30% or less. In addition, the First WHO International Reference Panel for anti-SARS-CoV-2 immunoglobulin (NIBSC, Code# 20/150) was continuously diluted for quantification purposes and quantified through ELISA analysis; neutralizing antibody values were quantified based on this.

### 2.6. Statistical Analysis

Through the survey, the subjects’ gender, age, height, weight, blood type, disease, type of vaccination, duration after inoculation, degree of pain, and preprocessing of data analysis (statistics) was performed using RStudio [27]. A Chi-square test or Fisher’s exact test was performed on the positive results of the POCT-type diagnostic kit in each vaccination subject and the collected information, and the significance of general and neutralizing antibody generation factors was verified. Independent sample t-test and variance analysis one-way ANOVA were performed to analyze the neutralization antibody production rate according to the factor of the neutralization antibody concentration value as a dependent variable in the ELISA method. Scheffe’s post hoc analysis was performed on variables showing significant differences. Multiple regression analysis was conducted to verify the effect of age, gender, vaccine type (classification group), pain, duration, blood type, and major chronic diseases on neutralizing antibody production.

## 3. Results

### 3.1. Clinical Study of RapiSure^TM^ (EDGC^TM^) COVID-19 S1 RBD IgG/Neutralizing Ab Test

For an evaluation of the POCT used in this study, clinical performance was evaluated on 264 clinical serum samples suitable for the subject and sample selection criteria. The samples used in this clinical performance evaluation were confirmed positive for neutralizing antibody production by PRNT_90_ among subjects whose COVID-19 infection was verified through RT-PCR. As a result of the evaluation of the S1 RBD IgG antibody, the clinical sensitivity was 96.8% (90/93) (95% confidence interval: 90.94% to 98.90%), and the clinical specificity was 97.7% (167/171) (95% confidence interval: 94.14% to 99.09%). As a result of the neutralizing antibody evaluation, the clinical sensitivity was 92.22% (83/90) (95% confidence interval: 84.81% to 96.18%), and the clinical specificity was 100.00% (174/174) (95% confidence interval: 97.84% to 100.00%) (Appendix A).

In this study, 736 people received one of five types of vaccines (AZD1222, BNT162b2, mRNA1273, AZD1222 + BNT162b2, Ad26COV2.S) and were subjected to a retrospective study on S1 RBD IgG antibody and neutralizing antibody production using a POCT. As a result of comparative analysis using the naked eye and measurement application (Appendix A) for each result, a 99% concordance rate was confirmed. Additionally, the results of the POCT obtained by measurements application interpretation showed a higher positive percent agreement (96.91% confidence interval: 94.97~98.12%) and negative percent agreement (97.60% confidence interval: 94.86~98.90%) with the ELISA test (Appendix A).

### 3.2. Analyzing Descriptive Statistics for Different Vaccinations

Table 1 shows the results of technical statistical analysis on significant variables according to vaccine type. In the case of mRNA1273 inoculators, the number of samples (*n* = 35) was small, and the results of both the COVID-19 S1 RBD IgG test and the neutralizing antibody test were positive. In terms of gender, 433 women (58.8%) and 303 men (41.2%) were recruited, and 10 women (21.3%) and 37 men (78.7%) were in the one-time inoculation group of the Ad26COV2.S vaccine, indicating that there was no ratio difference in the rest of the group (*p* < 0.0001). In the case of age, the average age of subjects was 51.5 years (±15.2), the AZD1222 inoculation group had the highest average age of 63.4 years (±7.4), and the Ad26COV2.S inoculation group had the lowest average age of 45.7 years (±10.9). In the case of blood type, there is no significance in groups of samples (Table 1).

### 3.3. Analysis of Factors for Generating General Antibodies and Neutralizing Antibody after Vaccination

After vaccination, cross-samples were calculated for factor analysis on the generation of the S1RBD IgG antibody and neutralizing antibody, and significance was verified by performing a Chi-square test and a Fisher’s exact test. In the case of the S1RBD IgG antibody, statistical significance was found regarding gender, age, type of vaccination vaccine, body mass index, diabetes, post-vaccine pain, and post-vaccine progress variables (*p* < 0.0001). In the neutralizing antibody, elements showing statistical significance overlapped with those of the S1RBD IgG antibody, but there was no statistical significance found regarding gender (*p* = 0.21), and it was significant in HYPERLIPIDEMIA (HiBP) (*p* = 0.0003). In addition, there was no statistical significance in blood type (*p* = 0.95) (Table 2).

### 3.4. Analysis of Neutralizing Antibody Production Rate According to Various Factors

Variance analysis (mean, equal variance assumption satisfaction, median, and equal variance assumption dissatisfaction) was performed to verify the significance of the major variables of age, gender, vaccine type (classification group), pain, duration, blood type, and major chronic diseases on neutralization antibody production. As a result of the analysis, significant differences were found regarding gender (*p* = 0.021), age (*p* < 0.0001), vaccine type (*p* < 0.0001), vaccination interval (*p* < 0.0001), and pain score (*p* < 0.0001). In addition, among those with chronic diseases, a significant difference was found between those with diabetes (*p* < 0.0001), high blood pressure (*p* = 0.002), and chronic fatigue (*p* = 0.018) (Table 3).

As a result of conducting a follow-up analysis of Sheppe’s variables showing significant differences, mRNA-based vaccines, cross-vaccination (AZD1222 + BNT162b2), AZD1222, and Ad26COV2.S showed higher rates of neutralizing antibody production (Table 3), and the age-specific neutralizing antibodies were higher in their 30 s, 50 s, and 50 s. In addition, in all vaccines except AZD1222, it was confirmed that the rate of neutralizing antibody production decreased statistically significantly when more than 90 days passed after vaccination (*p* < 0.05), and in all vaccines, the rate of neutralizing antibody production was significant according to pain after vaccination (Table 4).

### 3.5. Verification of Antibody Production Rate through Multiple Regression Analysis

Multiple regression analysis was conducted to verify the effect on the production rate of neutralizing antibody by each factor, such as age and major chronic diseases. Gender, vaccine type, blood type, and major chronic diseases were dummy converted and treated as control variables. The regression model was statistically significant (*F* = 50.70379, *p* < 0.001), and the explanatory power was 59.5% (R^2^ 59.5%/R^2^ adjusted 58.4%). As a result of verifying the significance of the regression coefficient, the results of AZD1222 + BNT162b2 (β = 728.55, *p* < 0.001), BNT162b (β = 814.33, *p* < 0.001), and mRNA-1273 (β = 978.87, *p* < 0.001) were confirmed, and the degree of pain after vaccination (β = 104.92, *p* = 0.001) was affected in a significant direction. The neutralizing antibody production rate was high for AZD1222 + BNT162b2 and the mRNA vaccine (*p* < 0.001), and Ad26.COV2.S showed a low neutralizing antibody production rate compared to AZD1222 (*p* < 0.001) (Figure 1). In all vaccine groups, the production rate of neutralizing antibody decreased as age increased (β = −8.05, *p* < 0.001) (Figure 2), and in all vaccines except AZD1222, the production rate of neutralizing antibody decreased statistically significantly after 90 days or more after vaccination (*p* < 0.05) (Figure 3). It was shown that the more severe the pain after all vaccinations, the higher the rate of neutralizing antibody production (Figure 4). For all vaccines, subjects with diabetes (β = −282.23, *p* = 0.001) showed lower rates of neutralizing antibody production after inoculation (Table 4).

## 4. Discussion

We developed a new diagnostic POCT that quickly and conveniently diagnoses SARS-CoV2-2 S1 RBD IgG antibody and neutralizing antibody in whole blood, serum, and plasma. In clinical performance (264 serum confirmed positive for neutralizing antibody production by PRNT_90_), the S1 RBD IgG antibody showed 96.8% of sensitivity and 97.7% of specificity. In the case of neutralizing antibody, 92.22% of sensitivity and 100.00% of specificity were evaluated (Appendix A). The diagnostic performance of the test by measurement application for interpretation, which is a 99% concordance rate, was compared with naked eye evaluation (Appendix A). As shown in Appendix A, the newly developed POCT showed excellent performance comparable to that of the ELISA, although the sensitivity was slightly lower. In the interpretation of POCT, visual reading can make interpretation errors, so research is focused on making accurate and improved measurements through interpretation devices [28,29]. When the signal intensity of neutralizing antibody was measured using an application, the T/R ratio cutoff increased in the sensitivity of the POCT. According to the results, it may be confirmed that POCT showed a higher correlation with the ELISA than the PRNT_90_. It is reported that the ELISA test specificities can be variable depending on the used reference standard [30]. In our newly developed POCT, we use only the purified RBD portion of spike protein, similar to the ELISA kit assay principle. It is reported that the principal neutralizing domain is the N-terminal domain of the spike protein RBD portion [15,16]. It is thought that various changes in the tertiary structure of the virus’s spike protein in vivo may occur, which is expected to cause changes in the recognition site of the epitope recognized by neutralizing antibody. For this reason, it is expected that there may be a difference between the ELISA and the PRNT_90_ [31]. So, we need further investigation of the performance of POCT, especially its sensitivity.

Due to the high mortality and transmission power of SARS-CoV-2 diseases, the negative aspects of SARS-CoV-2 antibody POCT testing have been emphasized due to the relatively high false positive and false negative results in early infections because of varying degrees of performance. However, as specific data on SARS-CoV-2 antibody POCTs are accumulated, the usefulness of antibody tests is increased for the confirmation of recent/previous infections, confirmation of effectiveness after vaccination, and serological prevalence investigation. However, there are still many areas that need to be revealed in SARS-CoV-2 antibody POCTs, such as which antigen, which epitope site is the ideal target site, duration time by antibody type, correlation between antibody production and immunity, differences in immune response according to vaccine type, and deformation. Nevertheless, if specific data and knowledge about them are accumulated, the advantages and limitations of individual SARS-CoV-2 antibody POCTs become clear, and SARS-CoV-2 antibody tests are expected to be more useful in SARS-CoV2 treatment procedure [32].

Since the outbreak of COVID-19, more than ten types of vaccines have been developed and used worldwide, but the preventive effect varies. Differences in the formation of neutralizing antibodies according to individuals after vaccination and the persistence of neutralizing antibodies are important factors in prevention and are considered important criteria for determining the timing and necessity of additional vaccination. Neutralizing antibodies prevent the binding of the S1-RBD spike protein of COVID19 to the ACE2 receptor in the respiratory mucous membrane, the path through which the COVID-19 virus penetrates the human body. During the epidemic, vaccines are the most effective method of preventing the spread of the virus. Therefore, the generation or maintenance of neutralizing antibody has been used as important experimental indicators in the development of vaccines. The production of neutralizing antibody varies greatly depending on the type of vaccine; this was reported based on the clinical trials of seven vaccines regarding neutralizing antibody production (i.e., prevention of coronavirus) [33]. In this study, the production of neutralizing antibodies were the highest for mRNA-1273 and BNT162b2, which are the mRNA vaccines, followed by AZD1222 + BNT162b2 (cross-inoculation), AZD1222, and Ad26.COV2.S (Table 2 and Table 3, Figure 1). Antibodies were generated in the same order for neutralizing antibody and immunoglobulin (IgG).

Age is one of the most critical factors affecting the production of immunoglobulin and neutralizing antibody. The production of neutralizing antibody by age after two doses of BNT162b vaccine administration showed that the neutralizing value of neutralizing antibody decreased proportionally with age [34,35]. This study also showed that the neutralizing antibody decreased with age regardless of the type of vaccine. However, more men in their 30 s received the Ad26.COV2.S vaccine compared to those in other age groups, so the production rate of neutralizing antibody was slightly lower than that of those in their 40 s who mainly received the BNT162b vaccine (Figure 2). When other factors such as vaccine type were controlled in the multiple regression analysis, it was found that age was a critical factor in neutralization antibody production (*p* < 0.001, Table 4). The production of S1 RBD IgG occurs at around two weeks post onset of vaccination, and most people have neutralizing titers from two weeks to three months, with excellent titer variability. Therefore, neutralizing antibody production positively correlates with age. In general, older people have reduced immune responses to infection and vaccination. B cell activation is vital for the efficacy of antibody production, but there are several age-related changes in functions and phenotypes in B cells. We hypothesize that some of these changes produce neutralizing antibody production levels [36,37].

Gender is also one of the most important factors related to the production of neutralizing antibodies; in this study, there were higher production rates in men than in women (Table 3). In general, there was a tendency for the antibody values to be higher in men than in women [38,39]; however, there was no statistical significance in the multiple regression analysis (Table 4). Additionally, blood type and body mass index did not affect the production of immunoglobulin and neutralizing antibody.

Personal medical history or immune status were determined to affect the production of general and neutralizing antibodies. Individual diseases and symptoms were investigated together through a questionnaire. As a result, there was a statistical difference among those with hypertension (neutral antibody *p* = 0.009), diabetes (neutral antibody *p* < 0.001, general antibody *p* = 0.008), hyperlipidemia (neutral antibody *p* = 0.003), cancer history (neutral antibody *p* = 0.041), and chronic fatigue syndrome (neutral antibody *p* = 0.012). There was no statistically significant difference among those with insomnia, irritable bowel syndrome, and subjective immunosuppression (Table 2).

As a result of the multiple regression analysis, it was found that major variables such as vaccine type, gender, and age were controlled, and in the case of those with diabetes, neutralizing antibody production was less likely (Table 4). In addition, it is reported that the higher blood sugar in diabetes patients, the lower the immunity to the COVID-19 vaccine, resulting in less neutralization antibody production, so it is vital to control blood sugar in diabetes patients to produce and maintain antibodies [40].

The duration of neutralizing antibody after the last vaccination has a significant meaning from a quarantine point of view. Unfortunately, for most vaccines, neutralizing antibodies do not last long after vaccine administration. For example, according to a study published by Widge et al. in 2021 [39], the peak of the neutralizing antibody value is about 43 days after two doses of mRNA-1273 inoculation, regardless of age, and gradually decreases after that. In this study, the general antibody production rates fell proportionally to 97.3% within one month, 91.0% within three months, and 75.2% after three months of administrating the vaccine, regardless of vaccine type (Table 2). In particular, in the case of mRNA vaccines BNT16b2 and mRNA-1273, there were significant differences in the neutralizing antibody production rates around the three months (Figure 3).

Finally, as a result of analyzing the pain score and antibody production at the time of vaccination, the more severe the pain score, the higher the likelihood of both general and neutralizing antibody generation (Table 2). This result was also statistically significant in the multiple regression analysis in which all other factors were controlled (*p* = 0.001, Table 4). This result is opposed to another study [41,42], which examined the relationship between pain and antibody production in 206 healthy workers and did not correlate pain with neutralizing antibodies after BNT16b2 administration.

With the limitation of POCT, it is necessary to confirm the (1) continuous antibody status change monitoring, (2) sensitivity analysis according to antibody epidemiology, (3) limited detection of sensitivity, and (4) difference in sensitivity analysis due to infection and vaccination. Additionally, types of vaccines were forcibly administered to specific age groups according to The Korea Disease Control and Prevention Agency (KDCA) policy, so different types of vaccines were not evenly distributed, and different age groups had varying results. However, this study is meaningful as the production rate of immunoglobulin and neutralizing antibody for various types of vaccines can be directly compared. In particular, there are not many studies on cross-vaccination worldwide, and a few subjects who were vaccinated with AZD1222 + BNT162b in Korea participated in this study. The production rates of both immunoglobulin and neutralizing antibody were significantly higher than for AZD1222 alone.

## 5. Conclusions

The clinical use of POCT is possible by showing the same results for major factors such as vaccine type, age, elapsed after vaccine administration, diabetes, and post-vaccine pain. The findings of this study are significant in providing essential points of reference for future vaccine policies.

## Figures and Tables

**Figure 1 diagnostics-12-01924-f001:**
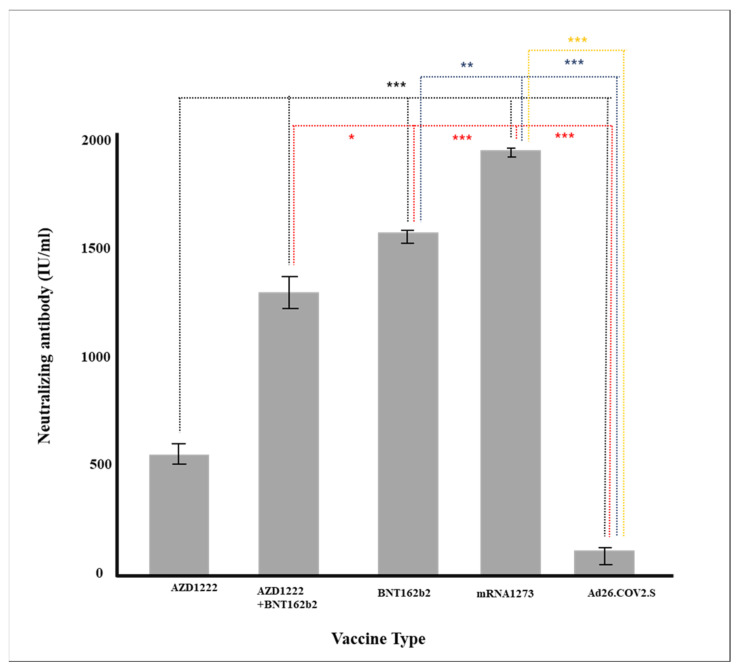
Production of neutralizing antibody according to vaccine (* *p* < 0.05, ** *p* < 0.01, *** *p* < 0.001).

**Figure 2 diagnostics-12-01924-f002:**
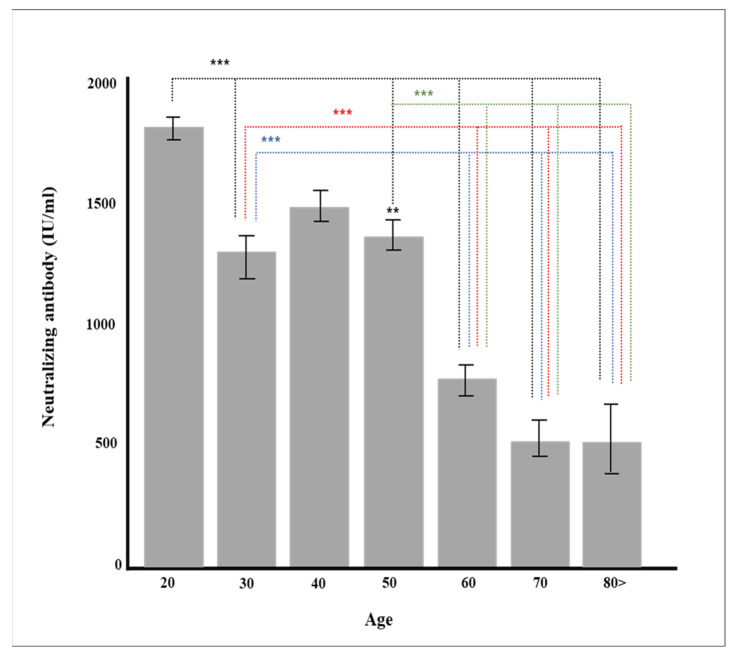
Production of neutralizing antibody according to age (** *p* < 0.01, *** *p* < 0.001).

**Figure 3 diagnostics-12-01924-f003:**
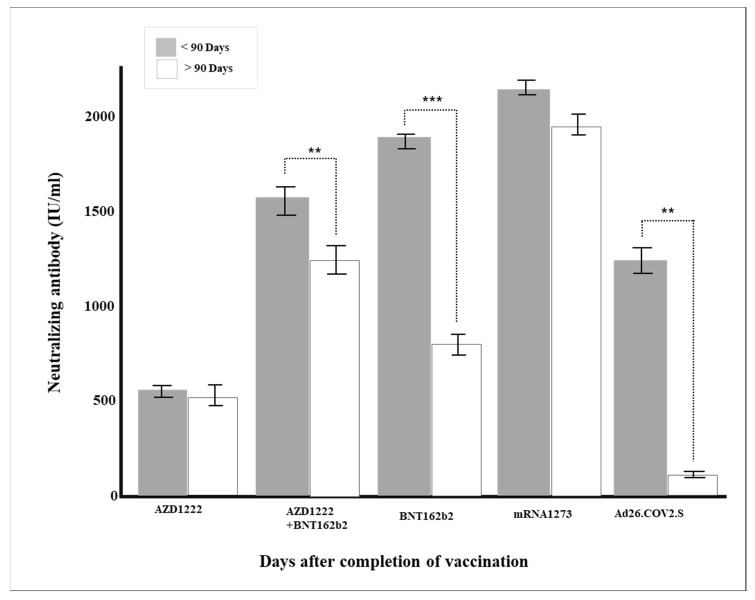
Changes in neutralizing antibody production according to the period after vaccination (** *p* < 0.01, *** *p* < 0.001).

**Figure 4 diagnostics-12-01924-f004:**
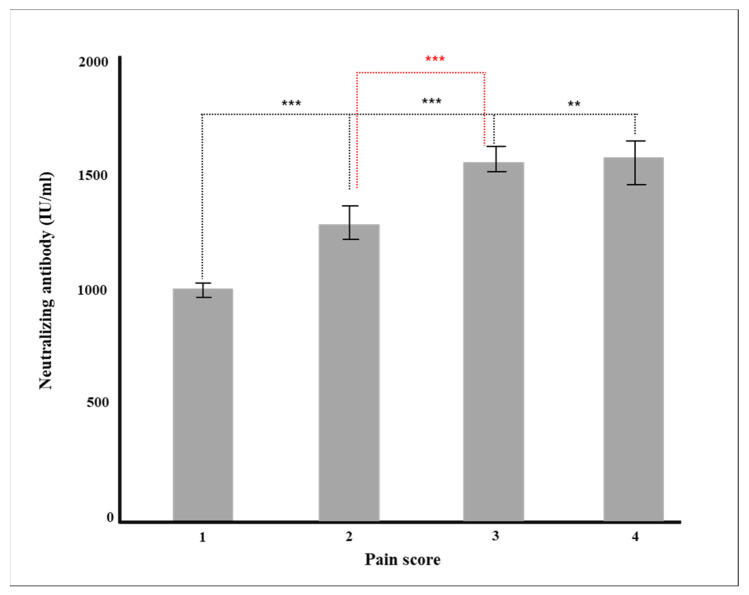
Production of neutralizing antibody according to pain after vaccination (** *p* < 0.01, *** *p* < 0.001).

**Table 1 diagnostics-12-01924-t001:** Subject recruitment criteria in the study.

	Total	AZD1222	AZD1222 + BNT162b2	BNT162b2	mRNA-1273	Ad26.COV2.S	*p*-Value
**Sex**							**<0.0001**
F	433 (58.8%)	137 (66.8%)	52 (55.9%)	216 (61.2%)	18 (47.4%)	10 (21.3%)	
M	303 (41.2%)	68 (33.2%)	41 (44.1%)	137 (38.8%)	20 (52.6%)	37 (78.7%)	
**AGE**	51.5 (±15.2)	63.4 (±7.4)	46.0 (±9.7)	47.9 (±16.7)	41.4 (±10.8)	45.7 (±10.9)	**<0.0001**
**Age Group**							**<0.0001**
20’s	74 (10.1%)	0 (0.0%)	0 (0.0%)	67 (19.0%)	7 (18.4%)	0 (0.0%)	
30’s	105 (14.3%)	3 (1.5%)	22 (23.7%)	51 (14.4%)	9 (23.7%)	20 (42.6%)	
40’s	137 (18.6%)	6 (2.9%)	44 (47.3%)	65 (18.4%)	13 (34.2%)	9 (19.1%)	
50’s	171 (23.2%)	35 (17.1%)	18 (19.4%)	99 (28.0%)	7 (18.4%)	12 (25.5%)	
60’s	162 (22.0%)	124 (60.5%)	6 (6.5%)	26 (7.4%)	2 (5.3%)	4 (8.5%)	
70’s	72 (9.8%)	37 (18.0%)	3 (3.2%)	30 (8.5%)	0 (0.0%)	2 (4.3%)	
Over 80’s	15 (2.0%)	0 (0.0%)	0 (0.0%)	15 (4.2%)	0 (0.0%)	0 (0.0%)	
**Bloodtype**							**0.69**
A	282 (38.3%)	79 (38.5%)	33 (35.5%)	141 (39.9%)	13 (34.2%)	16 (34.0%)	
B	180 (24.5%)	54 (26.3%)	22 (23.7%)	76 (21.5%)	14 (36.8%)	14 (29.8%)	
O	207 (28.1%)	54 (26.3%)	27 (29.0%)	107 (30.3%)	8 (21.1%)	11 (23.4%)	
AB	67 (9.1%)	18 (8.8%)	11 (11.8%)	29 (8.2%)	3 (7.9%)	6 (12.8%)	
**BMI**	23.1 (±3.4)	23.2 (±2.7)	22.9 (±3.4)	23.0 (±3.6)	22.6 (±3.4)	24.4 (±4.0)	**0.075**
**BMI Group**							**0.001**
Underweight	37 (5.0%)	5 (2.4%)	7 (7.5%)	21 (5.9%)	4 (10.5%)	0 (0.0%)	
Normal	343 (46.6%)	89 (43.4%)	44 (47.3%)	172 (48.7%)	16 (42.1%)	22 (46.8%)	
Overweight	161 (21.9%)	63 (30.7%)	14 (15.1%)	72 (20.4%)	8 (21.1%)	4 (8.5%)	
Obesity	168 (22.8%)	44 (21.5%)	27 (29.0%)	71 (20.1%)	9 (23.7%)	17 (36.2%)	
High obesity	27 (3.7%)	4 (2.0%)	1 (1.1%)	17 (4.8%)	1 (2.6%)	4 (8.5%)	
**HBP**							**0.0002**
NO	611 (83.0%)	151 (73.7%)	86 (92.5%)	297 (84.1%)	35 (92.1%)	42 (89.4%)	
YES	125 (17.0%)	54 (26.3%)	7 (7.5%)	56 (15.9%)	3 (7.9%)	5 (10.6%)	
**Diabetes**							**0.074**
NO	688 (93.5%)	183 (89.3%)	90 (96.8%)	334 (94.6%)	37 (97.4%)	44 (93.6%)	
YES	48 (6.5%)	22 (10.7%)	3 (3.2%)	19 (5.4%)	1 (2.6%)	3 (6.4%)	
**Hyperlipidemia**							**<0.0001**
NO	586 (79.6%)	134 (65.4%)	78 (83.9%)	296 (83.9%)	34 (89.5%)	44 (93.6%)	
YES	150 (20.4%)	71 (34.6%)	15 (16.1%)	57 (16.1%)	4 (10.5%)	3 (6.4%)	
**History of Cancer**							**0.14**
NO	688 (93.5%)	188 (91.7%)	90 (96.8%)	328 (92.9%)	35 (92.1%)	47 (100.0%)	
YES	48 (6.5%)	17 (8.3%)	3 (3.2%)	25 (7.1%)	3 (7.9%)	0 (0.0%)	
**Chronic Fatigue**							**0.13**
NO	693 (94.2%)	194 (94.6%)	83 (89.2%)	333 (94.3%)	36 (94.7%)	47 (100.0%)	
YES	43 (5.8%)	11 (5.4%)	10 (10.8%)	20 (5.7%)	2 (5.3%)	0 (0.0%)	
**Insomnia**							**0.048**
NO	662 (89.9%)	173 (84.4%)	86 (92.5%)	323 (91.5%)	35 (92.1%)	45 (95.7%)	
YES	74 (10.1%)	32 (15.6%)	7 (7.5%)	30 (8.5%)	3 (7.9%)	2 (4.3%)	
**IBS**							**0.80**
NO	692 (94.0%)	190 (92.7%)	89 (95.7%)	333 (94.3%)	35 (92.1%)	45 (95.7%)	
YES	44 (6.0%)	15 (7.3%)	4 (4.3%)	20 (5.7%)	3 (7.9%)	2 (4.3%)	
**Weakened Immunity**							**0.69**
NO	695 (94.4%)	193 (94.1%)	89 (95.7%)	334 (94.6%)	34 (89.5%)	45 (95.7%)	
YES	41 (5.6%)	12 (5.9%)	4 (4.3%)	19 (5.4%)	4 (10.5%)	2 (4.3%)	
**Degree of Pain after Vaccination**	1.6 (±0.7)	1.5 (±0.6)	1.8 (±0.6)	1.6 (±0.7)	2.1 (±0.8)	1.7 (±0.9)	**<0.0001**
**Degree of Pain after Vaccination (Ordinal Data)**							**<0.0001**
1	321 (43.6%)	106 (51.7%)	21 (22.6%)	162 (45.9%)	6 (15.8%)	26 (55.3%)	
1.5	112 (15.2%)	39 (19.0%)	17 (18.3%)	50 (14.2%)	6 (15.8%)	0 (0.0%)	
2	174 (23.6%)	34 (16.6%)	34 (36.6%)	84 (23.8%)	10 (26.3%)	12 (25.5%)	
2.5	77 (10.5%)	16 (7.8%)	15 (16.1%)	37 (10.5%)	9 (23.7%)	0 (0.0%)	
3	36 (4.9%)	7 (3.4%)	3 (3.2%)	16 (4.5%)	4 (10.5%)	6 (12.8%)	
3.5	7 (1.0%)	1 (0.5%)	2 (2.2%)	3 (0.8%)	1 (2.6%)	0 (0.0%)	
4	9 (1.2%)	2 (1.0%)	1 (1.1%)	1 (0.3%)	2 (5.3%)	3 (6.4%)	
**Days after Completion of Vaccination**	77.1 (±41.9)	76.4 (±31.8)	100.8 (±32.4)	67.7 (±43.0)	41.8 (±29.6)	132.4 (±24.2)	**<0.0001**
**Days after Completion of Vaccination (Ordinal Data)**							**<0.0001**
~30 days	74 (10.1%)	5 (2.4%)	1 (1.1%)	54 (15.3%)	14 (36.8%)	0 (0.0%)	
30~90 days	420 (57.1%)	140 (68.3%)	36 (38.7%)	220 (62.3%)	22 (57.9%)	2 (4.3%)	
After 90 days	242 (32.9%)	60 (29.3%)	56 (60.2%)	79 (22.4%)	2 (5.3%)	45 (95.7%)	

**Table 2 diagnostics-12-01924-t002:** Analysis of general antibody and neutralizing antibody production rate according to various factors.

	Neutralizing Antibody				S1 RBD IgG Antibody			
	Total	Negative	Positive	*p*-Value	Total	Negative	Positive	*p*-Value
**Sex**								
F	433 (58.8%)	139 (32.1%)	294 (67.9%)	**0.21**	433 (58.8%)	37 (8.5%)	396 (91.5%)	**<0.0001**
M	303 (41.2%)	111 (36.6%)	192 (63.4%)		303 (41.2%)	63 (20.8%)	240 (79.2%)	
**Age**	51.5 (±15.2)	59.6 (±13.4)	47.3 (±14.4)	**<0.0001**	51.5 (±15.2)	58.0 (±13.7)	50.5 (±15.2)	**<0.0001**
**Age Group**								
20’s	74 (10.1%)	1 (1.4%)	73 (98.6%)	**<0.0001**	74 (10.1%)	0 (0.0%)	74 (100.0%)	**<0.0001**
30’s	105 (14.3%)	30 (28.6%)	75 (71.4%)		105 (14.3%)	18 (17.1%)	87 (82.9%)	
40’s	137 (18.6%)	28 (20.4%)	109 (79.6%)		137 (18.6%)	10 (7.3%)	127 (92.7%)	
50’s	171 (23.2%)	41 (24.0%)	130 (76.0%)		171 (23.2%)	16 (9.4%)	155 (90.6%)	
60’s	162 (22.0%)	93 (57.4%)	69 (42.6%)		162 (22.0%)	38 (23.5%)	124 (76.5%)	
70’s	72 (9.8%)	47 (65.3%)	25 (34.7%)		72 (9.8%)	14 (19.4%)	58 (80.6%)	
Over 80’s	15 (2.0%)	10 (66.7%)	5 (33.3%)		15 (2.0%)	4 (26.7%)	11 (73.3%)	
**Bloodtype**								
A	282 (38.3%)	96 (34.0%)	186 (66.0%)	**0.95**	282 (38.3%)	33 (11.7%)	249 (88.3%)	**0.36**
B	180 (24.5%)	64 (35.6%)	116 (64.4%)		180 (24.5%)	29 (16.1%)	151 (83.9%)	
O	207 (28.1%)	68 (32.9%)	139 (67.1%)		207 (28.1%)	26 (12.6%)	181 (87.4%)	
AB	67 (9.1%)	22 (32.8%)	45 (67.2%)		67 (9.1%)	12 (17.9%)	55 (82.1%)	
**BMI**	23.1 (±3.4)	23.3 (±3.0)	23.0 (±3.6)	**0.046**	23.1 (±3.4)	23.7 (±3.1)	23.0 (±3.4)	**0.013**
**BMI Group**								
Underweight	37 (5.0%)	11 (29.7%)	26 (70.3%)	**0.17**	37 (5.0%)	5 (13.5%)	32 (86.5%)	**0.25**
Normal	343 (46.6%)	106 (30.9%)	237 (69.1%)		343 (46.6%)	40 (11.7%)	303 (88.3%)	
Overweight	161 (21.9%)	62 (38.5%)	99 (61.5%)		161 (21.9%)	20 (12.4%)	141 (87.6%)	
Obesity	168 (22.8%)	65 (38.7%)	103 (61.3%)		168 (22.8%)	32 (19.0%)	136 (81.0%)	
High obesity	27 (3.7%)	6 (22.2%)	21 (77.8%)		27 (3.7%)	3 (11.1%)	24 (88.9%)	
**HBP**								
No	611 (83.0%)	191 (31.3%)	420 (68.7%)	**0.0009**	611 (83.0%)	80 (13.1%)	531 (86.9%)	**0.39**
Yes	125 (17.0%)	59 (47.2%)	66 (52.8%)		125 (17.0%)	20 (16.0%)	105 (84.0%)	
**Diabetes**								
No	688 (93.5%)	220 (32.0%)	468 (68.0%)	**<0.0001**	688 (93.5%)	87 ( 12.6%)	601 (87.4%)	**0.008**
Yes	48 (6.5%)	30 (62.5%)	18 (37.5%)		48 (6.5%)	13 (27.1%)	35 (72.9%)	
**Hyperlipidemia**								
No	586 (79.6%)	180 (30.7%)	406 (69.3%)	**0.0003**	586 (79.6%)	76 (13.0%)	510 (87.0%)	**0.35**
Yes	150 (20.4%)	70 (46.7%)	80 (53.3%)		150 (20.4%)	24 (16.0%)	126 (84.0%)	
**History of cancer**								
No	688 (93.5%)	227 (33.0%)	461 (67.0%)	**0.041**	688 (93.5%)	92 (13.4%)	596 (86.6%)	**0.51**
Yes	48 (6.5%)	23 (47.9%)	25 (52.1%)		48 (6.5%)	8 (16.7%)	40 (83.3%)	
**Chronic fatigue**								
No	693 (94.2%)	243 (35.1%)	450 (64.9%)	**0.012**	693 (94.2%)	97 (14.0%)	596 (86.0%)	**0.25**
Yes	43 ( 5.8%)	7 (16.3%)	36 (83.7%)		43 (5.8%)	3 (7.0%)	40 (93.0%)	
**Insomnia**								
No	662 (89.9%)	219 (33.1%)	443 (66.9%)	**0.15**	662 (89.9%)	93 (14.0%)	569 (86.0%)	**0.37**
Yes	74 (10.1%)	31 (41.9%)	43 (58.1%)		74 (10.1%)	7 (9.5%)	67 (90.5%)	
**IBS**								
No	692 (94.0%)	237 (34.2%)	455 (65.8%)	**0.62**	692 (94.0%)	91 (13.2%)	601 (86.8%)	**0.17**
Yes	44 (6.0%)	13 (29.5%)	31 (70.5%)		44 (6.0%)	9 ( 20.5%)	35 (79.5%)	
**Weakened immunity**								
No	695 (94.4%)	234 (33.7%)	461 (66.3%)	**0.5**	695 (94.4%)	95 (13.7%)	600 (86.3%)	**1**
Yes	41 (5.6%)	16 (39.0%)	25 (61.0%)		41 (5.6%)	5 (12.2%)	36 (87.8%)	
**Degree of Pain after Vaccination**	1.6 (±0.7)	1.5 (±0.7)	1.7 (±0.7)	**<0.0001**	1.6 (±0.7)	1.4 (±0.7)	1.7 (±0.7)	**<0.0001**
**Degree of Pain after Vaccination**								
1	321 (43.6%)	144 (44.9%)	177 (55.1%)	**<0.0001**	321 (43.6%)	64 (19.9%)	257 (80.1%)	**<0.0001**
1.5	112 (15.2%)	38 (33.9%)	74 (66.1%)		112 (15.2%)	11 (9.8%)	101 (90.2%)	
2	174 (23.6%)	41 (23.6%)	133 (76.4%)		174 (23.6%)	15 (8.6%)	159 (91.4%)	
2.5	77 (10.5%)	8 (10.4%)	69 (89.6%)		77 (10.5%)	1 (1.3%)	76 (98.7%)	
3	36 (4.9%)	15 (41.7%)	21 (58.3%)		36 (4.9%)	6 (16.7%)	30 (83.3%)	
3.5	7 (1.0%)	0 (0.0%)	7 (100.0%)		7 (1.0%)	0 (0.0%)	7 (100.0%)	
4	9 (1.2%)	4 (44.4%)	5 (55.6%)		9 (1.2%)	3 (33.3%)	6 (66.7%)	
**Days after completion of Vaccination**	77.1 (±41.9)	99.3 (±40.1)	65.7 (±38.0)	**<0.0001**	77.1 (±41.9)	104.9 (±42.6)	72.7 (±40.0)	**<0.0001**
**Days after completion of Vaccination**								
~30 days	74 (10.1%)	2 (2.7%)	72 (97.3%)	**<0.0001**	74 (10.1%)	2 (2.7%)	72 (97.3%)	**<0.0001**
30~90 days	420 (57.1%)	110 (26.2%)	310 (73.8%)		420 (57.1%)	38 (9.0%)	382 (91.0%)	
After 90 days	242 (32.9%)	138 (57.0%)	104 (43.0%)		242 (32.9%)	60 (24.8%)	182 (75.2%)	
**Classification**								
AZD1222	205 (27.9%)	135 (65.9%)	70 (34.1%)	**<0.0001**	205 (27.9%)	59 (28.8%)	146 (71.2%)	**<0.0001**
AZD1222 + BNT162b2	93 (12.6%)	24 (25.8%)	69 (74.2%)		93 (12.6%)	3 (3.2%)	90 (96.8%)	
BNT162b2	353 (48.0%)	49 (13.9%)	304 (86.1%)		353 (48.0%)	8 (2.0%)	345 (97.7%)	
mRNA-1273	38 (5.2%)	0 (0.0%)	38 (100.0%)		38 (5.2%)	0 (0.0%)	38 (100.0%)	
Ad26.COV2.S	47 (6.4%)	42 (89.4%)	5 (10.6%)		47 (6.4%)	30 (63.8%)	17 (36.2%)	

**Table 3 diagnostics-12-01924-t003:** Production of neutralizing antibody according to vaccine type, pain, duration, blood type, and major chronic disease.

		Count	PRNT (IU/mL) Mean (SD)
**Sex**	Total	675	1167.1 (±816.7)
	F	404	1216.2 (±780.5)
	M	271	1093.9 (±864.1)
	***p*-value**		**0.021**
**Age Group**	Total	675	1167.1 (±816.7)
	20’s	59	1807.7 (±351.0)
	30’s	91	1272.2 (±849.4)
	40’s	120	1489.5 (±718.7)
	50’s	161	1368.5 (±775.3)
	60’s	157	768.2 (±730.2)
	70’s	72	525.6 (±619.4)
	Over 80’s	15	523.3 (±557.7)
	***p*-value**		**<0.0001**
**Blood Type**	Total	675	1167.1 (±816.7)
	A	258	1191.3 (±817.9)
	B	157	1084.3 (±812.1)
	O	196	1212.0 (±815.5)
	AB	64	1134.9 (±829.6)
	***p*-value**		**0.47**
**BMI Group**	Total	675	1167.1 (±816.7)
	Underweight	31	1347.4 (±779.6)
	Normal	322	1227.3 (±777.6)
	Overweight	150	1096.6 (±815.9)
	Obesity	147	1046.0 (±892.9)
	High obesity	25	1303.4 (±817.9)
	***p*-value**		**0.121**
**Classification**	Total	675	1167.1 (±816.7)
	AZD1222	198	572.9 (±629.3)
	AZD1222 + BNT162b2	89	1339.4 (±706.4)
	BNT162b2	310	1566.2 (±639.1)
	mRNA-1273	33	992.2 (±103.7)
	Ad26.COV2.S	45	86.2 (±304.5)
	***p*-value**		**<0.0001**
**Days after Completion of Vaccination**	Total	675	1167.1 (±816.7)
	~30 DAYS	71	1861.2 (±475.8)
	30~90 DAYS	383	1324.4 (±769.5)
	90~ DAYS	221	671.4 (±705.6)
	***p*-value**		**<0.0001**
**Degree of Pain after Vaccination**	Total	675	1167.1 (±816.7)
	1	296	948.3 (±822.4)
	1.5	104	1231.4 (±795.7)
	2	162	1326.1 (±784.3)
	2.5	66	1600.5 (±588.3)
	3	33	1202.4 (±827.0)
	3.5	6	1878.1 (±175.6)
	4	8	950.9 (±980.9)
	***p*-value**		**<0.0001**
**Diabetes**	Total	675	1167.1 (±816.7)
	NO	628	1212.0 (±805.4)
	YES	47	566.9 (±731.8)
	***p*-value**		**<0.0001**
**HBP**	Total	675	1167.1 (±816.7)
	NO	528	1218.9 (±803.8)
	YES	147	981.1 (±838.0)
	***p*-value**		**0.002**
**History of Cancer**	Total	675	1167.1 (±816.7)
	NO	629	1178.1 (±815.2)
	YES	46	1017.3 (±830.9)
	***p*-value**		**0.21**
**Chronic Fatigue**	Total	675	1167.1 (±816.7)
	NO	635	1148.1 (±820.5)
	YES	40	1468.1 (±694.8)
	***p*-value**		**0.018**
**Insomnia**	Total	675	1167.1 (±816.7)
	NO	603	1186.2 (±822.4)
	YES	72	1006.8 (±753.6)
	***p*-value**		**0.062**
**IBS**	Total	675	1167.1 (±816.7)
	NO	635	1163.8 (±816.8)
	YES	40	1220.0 (±823.2)
	***p*-value**		**0.677**
**Weakened Immunity**	Total	675	1167.1 (±816.7)
	NO	635	1162.1 (±815.8)
	YES	40	1246.2 (±837.1)
	***p*-value**		**0.54**

**Table 4 diagnostics-12-01924-t004:** Multiple regression analysis of various factors according to neutralizing antibody production.

	PRNT (IU/mL)		
Predictors	Estimates	CI	*p*
**(Intercept)**	1380.67	979.52–1781.81	**<0.001**
**Sex (M, F = ref.)**	−63.44	−157.10–30.22	**0.184**
**Age**	−7.94	−11.55–−4.33	**<0.001**
**Classification (AZD1222 = ref.)**			
AZD1222 + BNT162b2	730.48	578.47–882.48	**<0.001**
BNT162b2	814.33	709.45–919.21	**<0.001**
mRNA-1273	978.87	769.62–1188.13	**<0.001**
Ad26.COV2.S	−286.57	−488.00–−85.14	**0.005**
**Blood Type (A = ref.)**			
B	21.06	−85.19–127.31	**0.697**
O	−19.00	−118.92–80.92	**0.709**
AB	−30.93	−176.70–114.84	**0.677**
**BMI**	3.42	−10.57–17.42	**0.631**
**HBP (YES, NO = ref.)**	−36.39	−157.71–84.93	**0.556**
**Diabetes (YES, NO = ref.)**	−283.29	−452.61–−113.97	**0.001**
**Hyperlipidemia (YES, NO = ref.)**	24.11	−81.76–129.98	**0.655**
**History of Cancer (YES, NO = ref.)**	−13.15	−178.13–151.82	**0.876**
**Chronic Fatigue (YES, NO = ref.)**	94.08	−93.21–281.37	**0.324**
**Insomnia (YES) (YES, NO = ref.)**	−30.86	−170.24–108.52	**0.664**
**IBS (YES, NO = ref.)**	46.92	−125.92–219.77	**0.594**
**Weakened Immunity (YES, NO = ref.)**	−98.44	−282.78–85.91	**0.295**
**Degree of Pain after vaccination**	97.15	34.63–159.66	**0.002**
**Days after Completion OF Vaccination**	−6.24	−7.40–−5.08	**<0.001**
Observations	675		
R^2^/R^2^ adjusted	0.597/0.585		

*F* = 48.43 (*p* < 0.001).

## Data Availability

Not applicable.

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
