# Peer review of "Analysis of Factors Affecting Neutralizing Antibody Production after COVID-19 Vaccination Using Newly Developed Rapid Point-of-Care Test"

_diagnostics, 2022, doi:10.3390/diagnostics12081924_

Round 1

Reviewer 1 Report

COMMENTS

In the present article the authors analyze the clinical performance of a novel POC LFIA that detects neutralizing SARS-COV-2 antibodies. Besides, and using a large sample set from a clinical retrospective study, the authors shed light on the antibody production (general antibody and neutralizing antibody) associated to current available vaccines by quantitatively measuring the antibodies using an ELISA kit. Different factors in antibody production are analyzed, and their statistical significance is assessed.

The authors bring a nice work that can have implications in future vaccine policies. However, and since the core of the article is based on the newly developed POC LFIA, it is striking that nothing is mentioned about the clinical performance of the LFIA under the Discussion section. The authors must explain the reason for this. Besides, and in page 9, it is mentioned that a 99% concordance rate with the ELISA kit is obtained in this retrospective clinical study. However, no data is shown respect this correlation. The authors must explain why.

The English language should be checked extensively. There are many sentences that are hard to follow, and many grammatical mistakes were detected. Example of a grammatical mistake. Page 10: “There was no statistical significance was found regarding blood type”.

How can be explained the high percentage of RBD IgG antibodies that the elder have but the relatively low percentage of neutralizing antibodies? It would be interesting to briefly hypothesize about these findings in the Discussion.

INTRODUCTION

Page 3. “The recently developed SARS-CoV-2 Immunoglobulin M (IgM)/Immunoglobulin G (IgG) antibody test can be utilized to confirm the production of antibodies (IgM) at the initial stage of infection and antibodies mainly involved in immune responses (IgG) after coronavirus infection” A reference is missing here. Which test are the authors referring to???

Page 4. Preclude instead of Avoid

Page 4. “Due to changes in the energetics between the mutated RBD and the neutralizing antibodies” instead of binding force.

Page 4.  MedRxiv references (refence 20) should not be used  

Page 4. Last paragraphs should be rewritten and should emphasize current Nab serology LFIAs such as doi: 10.1016/j.jcv.2021.105024.

Page 4-5. “Therefore, the development and application of POCT diagnostic devices using the same principle is considered an excellent method of overcoming limitations of conventional neutralizing antibody measurements [23].” Adding this reference is meaningless.

Page 5. The whole paragraph should be modified. A clearer explanation of the LFIA is necessary.

M&M

Page 6. Turbidity term in this context should be explained.

Page 6. IRB indicate that it is the acronym for IRB (Institutional Review Board)

Page 6. The correspondence between the vaccine ID and its common name should be shown somewhere. E.g., AZD1222 corresponds to Oxford-AstraZeneca vaccine.

Page 6. A picture or diagram of the Assay architecture is necessary. Does this correspond to the Figure S1? A complete description of the LFIA cassette and the two windows with all the TL, CL, RL is needed.

Page 7. Could not find info of the PRNT90 (IRB 2021GR0481, Korea University Guro Hospital) assay.

Page 8. Why RStudio has 3 references?

FIGURES

S1. Rewrite.

Table 1. Under hyperlipidemia it is written NO NO.

Table 2. Second column. A “t” is missing in Total.

How can be explained the high percentage of RBD IgG antibodies in the elder but low percentage of neutralizing antibodies?

Author Response

Re: Resubmission of manuscript “Analysis of Factors Affecting Neutralizing Antibody Production After COVID19 Vaccination”, Diagnostics-1841885

We would like to thank you for the letter and the opportunity to resubmit a revised copy of this manuscript. We would also like to take this opportunity to express our thanks to the reviewers for the positive feedback and helpful comments for correction or modification.

Following this letter are the editor and reviewer comments with our responses in italics, including how and where the text was modified. Changes made in the manuscript are marked using track changes.

The revision has been developed in consultation with all coauthors, and each author has given approval to the final form of this revision. The agreement form signed by each author remains valid.

We believe have resulted in an improved revised manuscript, which you will find uploaded alongside this document.

We very much hope the revised manuscript is accepted for publication in Journal.

Sincerely yours,

Gyu Ha Ryu.

on behalf of the authors

COMMENTS

In the present article the authors analyze the clinical performance of a novel POC LFIA that detects neutralizing SARS-COV-2 antibodies. Besides, and using a large sample set from a clinical retrospective study, the authors shed light on the antibody production (general antibody and neutralizing antibody) associated to current available vaccines by quantitatively measuring the antibodies using an ELISA kit. Different factors in antibody production are analyzed, and their statistical significance is assessed.

The authors bring a nice work that can have implications in future vaccine policies. However, and since the core of the article is based on the newly developed POC LFIA, it is striking that nothing is mentioned about the clinical performance of the LFIA under the Discussion section. The authors must explain the reason for this.

We appreciate your suggestions; we described the performance of the kit and our opinions in the discussion section.

Besides, and in page 9, it is mentioned that a 99% concordance rate with the ELISA kit is obtained in this retrospective clinical study. However, no data is shown respect this correlation. The authors must explain why.

We added the related data in Supplement tables 2 and 3 and described them in the result section.

The English language should be checked extensively. There are many sentences that are hard to follow, and many grammatical mistakes were detected. Example of a grammatical mistake. Page 10: “There was no statistical significance was found regarding blood type”.

We apologize for any inconvenience because the English expression is inappropriate. The broad wording and grammar of the manuscript were corrected.

How can be explained the high percentage of RBD IgG antibodies that the elder have but the relatively low percentage of neutralizing antibodies? It would be interesting to briefly hypothesize about these findings in the Discussion.

We suggested that issue in the discussion section.

INTRODUCTION

Page 3. “The recently developed SARS-CoV-2 Immunoglobulin M (IgM)/Immunoglobulin G (IgG) antibody test can be utilized to confirm the production of antibodies (IgM) at the initial stage of infection and antibodies mainly involved in immune responses (IgG) after coronavirus infection” A reference is missing here. Which test are the authors referring to

We added a corresponding reference. This does not mean a specific product but a general matter for the SARS-CoV-2 antibody POCT test.

Page 4. Preclude instead of Avoid

We changed the word as you suggested.

Page 4. “Due to changes in the energetics between the mutated RBD and the neutralizing antibodies” instead of binding force.

We changed the word as you suggested.

Page 4.  MedRxiv references (refence 20) should not be used  

We changed the reference as you suggested.

Page 4. Last paragraphs should be rewritten and should emphasize current Nab serology LFIAs such as doi: 10.1016/j.jcv.2021.105024.

We described the serological meaning of Nab in the introduction paragraph.

Page 4-5. “Therefore, the development and application of POCT diagnostic devices using the same principle is considered an excellent method of overcoming limitations of conventional neutralizing antibody measurements [23].” Adding this reference is meaningless.

Page 5. The whole paragraph should be modified. A clearer explanation of the LFIA is necessary.

We changed the last paragraph of the introduction and the reference.

M&M

Page 6. Turbidity term in this context should be explained.

We changed it to an appropriate expression.

Page 6. IRB indicate that it is the acronym for IRB (Institutional Review Board)

We spelled out the abbreviation for better understanding.

Page 6. The correspondence between the vaccine ID and its common name should be shown somewhere. E.g., AZD1222 corresponds to Oxford-AstraZeneca vaccine.

We described detailed vaccine information, including the full name of the vaccine and the company.

Page 6. A picture or diagram of the Assay architecture is necessary. Does this correspond to the Figure S1? A complete description of the LFIA cassette and the two windows with all the TL, CL, RL is needed.

We described the schematic diagram of our test and a more detailed description in Supplementary figure 1.

Page 7. Could not find info of the PRNT90 (IRB 2021GR0481, Korea University Guro Hospital) assay.

 We described the PRNT assay in more detail.

Page 8. Why RStudio has 3 references?

We kept the most relevant reference.

FIGURES

S1. Rewrite.

We rewrote the supplement in figure 1.

Table 1. Under hyperlipidemia it is written NO NO.

We corrected the word.

Table 2. Second column. A “t” is missing in Total.

We corrected the word.

How can be explained the high percentage of RBD IgG antibodies in the elder but low percentage of neutralizing antibodies?

We suggested our opinions in the discussion section.

Reviewer 2 Report

I had the pleasure of reviewing the work “Analysis of Factors Affecting Neutralizing Antibody Production After COVID19 Vaccination”. The work is not very innovative and the research group is relatively small (736 respondents), although it concerns a topic that is very important to all of us and I believe that each piece of information brings us closer to deepening the knowledge that gives us the opportunity to fight the COVID-19 pandemic. The work is well written. The introduction is exhaustive. Materials and methods as well as results are widely presented. The statistical methods were selected correctly. Relevant literature was used and appropriate conclusions were drawn. After the corrections, the work is worth publishing in Diagnostics.

Major revision:

1)    The format of the work is inconsistent with the guidelines for authors. There is also a lack of line numbering, which would facilitate the review. Please use the form of the Diagnostics journal.

2) In the abstract, we do not use abbreviations and their explanations. I believe that the abbreviation ELISA in the abstract is sufficient. In the introduction, please expand it.

3)    In Materials and methods, in the section "Human samples", please expand the abbreviations of each vaccine name. Full name, abbreviation and company producing the preparation.

4)    In Materials and Methods, in the subsection "Statistics", please describe what analysis of variance was used - one- or two-way ANOVA.

5)    I don't know exactly what the leitmotif of this manuscript is. Method or factors? The title suggests only factors.

6)  Descriptions of Figures 1-4 - For P <0.1 there should be two asterisks (**) and not three.

7)    Figures 1 and 4 - the gray bars overlap the x-axis.

8)  Figure S1 should be more elaborated. Photos should be aligned. It looks unsightly and careless.

9)    Description of Figure S1 - the last sentence "positive", please write in lowercase.

10) Discussion - The results should be discussed with more results from other scientists. There is a lot of work on this subject. Here the authors cite only one work on one issue. In my opinion, this is far too little.

Author Response

Re: Resubmission of manuscript “Analysis of Factors Affecting Neutralizing Antibody Production After COVID19 Vaccination”, Diagnostics-1841885

We would like to thank you for the letter and the opportunity to resubmit a revised copy of this manuscript. We would also like to take this opportunity to express our thanks to the reviewers for the positive feedback and helpful comments for correction or modification.

Following this letter are the editor and reviewer comments with our responses in italics, including how and where the text was modified. Changes made in the manuscript are marked using track changes.

The revision has been developed in consultation with all coauthors, and each author has given approval to the final form of this revision. The agreement form signed by each author remains valid.

We believe have resulted in an improved revised manuscript, which you will find uploaded alongside this document.

We very much hope the revised manuscript is accepted for publication in Journal.

Sincerely yours,

Gyu Ha Ryu.

on behalf of the authors

I had the pleasure of reviewing the work “Analysis of Factors Affecting Neutralizing Antibody Production After COVID19 Vaccination”. The work is not very innovative and the research group is relatively small (736 respondents), although it concerns a topic that is very important to all of us and I believe that each piece of information brings us closer to deepening the knowledge that gives us the opportunity to fight the COVID-19 pandemic. The work is well written. The introduction is exhaustive. Materials and methods as well as results are widely presented. The statistical methods were selected correctly. Relevant literature was used and appropriate conclusions were drawn. After the corrections, the work is worth publishing in Diagnostics.

Major revision:

1)    The format of the work is inconsistent with the guidelines for authors. There is also a lack of line numbering, which would facilitate the review. Please use the form of the Diagnostics journal.

We apologize for any inconvenience. We changed the format of the manuscript according to the journal of diagnostics template.

2) In the abstract, we do not use abbreviations and their explanations. I believe that the abbreviation ELISA in the abstract is sufficient. In the introduction, please expand it.

We spelled out the abbreviations in the abstract.

3)    In Materials and methods, in the section "Human samples", please expand the abbreviations of each vaccine name. Full name, abbreviation and company producing the preparation.

We described detailed vaccine information, including the full name of the vaccine and company.

4)    In Materials and Methods, in the subsection "Statistics", please describe what analysis of variance was used - one- or two-way ANOVA.

We described the accurate statical analysis method in the Statistics section.

5)    I don't know exactly what the leitmotif of this manuscript is. Method or factors? The title suggests only factors.

We changed the title as you suggested.

6)  Descriptions of Figures 1-4 - For P <0.1 there should be two asterisks (**) and not three.

We put the appropriate description, two asterisks (**).

7)    Figures 1 and 4 - the gray bars overlap the x-axis.

We corrected all figures the gray bars overlap the X-axis.

8)  Figure S1 should be more elaborated. Photos should be aligned. It looks unsightly and careless.

We improved Supplementary figure 1 and described our POCT test in more detail.

9)    Description of Figure S1 - the last sentence "positive", please write in lowercase.

We corrected the word.

10) Discussion - The results should be discussed with more results from other scientists. There is a lot of work on this subject. Here the authors cite only one work on one issue. In my opinion, this is far too little.

We described more discussion about the performance of our kit and our suggestions.

Round 2

Reviewer 2 Report

Thank you very much. The authors addressed all my comments and made appropriate corrections.